# Number of Local Regional Therapies for Hepatocellular Carcinoma and Peri-Operative Outcomes after Liver Transplantation

**DOI:** 10.3390/cancers15030620

**Published:** 2023-01-19

**Authors:** Audrey E. Brown, Amy M. Shui, Dieter Adelmann, Neil Mehta, Garrett R. Roll, Ryutaro Hirose, Shareef M. Syed

**Affiliations:** 1Department of Surgery, University of California, San Francisco, CA 94143, USA; 2Department of Epidemiology and Biostatistics, University of California, San Francisco, CA 94143, USA; 3Department of Anesthesia and Perioperative Care, University of California, San Francisco, San Francisco, CA 94143, USA; 4Department of Gastroenterology, University of California, San Francisco, CA 94143, USA; 5Department of Transplant Surgery, University of California, San Francisco, CA 94143, USA

**Keywords:** liver transplantation, hepatocellular carcinoma, local regional therapy, biliary complications, arterial complications

## Abstract

**Simple Summary:**

This was a retrospective single center analysis of 298 consecutive patients with HCC who underwent liver transplant. We sought to understand the impact that the number and type of local regional therapies (LRTs) have on peri-operative outcomes and complications after liver transplantation. The patients who received more than 3 LRTs prior to a liver transplant had a higher risk of biliary leak but otherwise similar outcomes to those who received fewer LRTs.

**Abstract:**

The wait times for patients with hepatocellular carcinoma (HCC) listed for liver transplant are longer than ever, which has led to an increased reliance on the use of pre-operative LRTs. The impact that multiple rounds of LRTs have on peri-operative outcomes following transplant is unknown. This was a retrospective single center analysis of 298 consecutive patients with HCC who underwent liver transplant (January 2017 to May 2021). The data was obtained from two institution-specific databases and the TransQIP database. Of the 298 patients, 27 (9.1%) underwent no LRTs, 156 (52.4%) underwent 1-2 LRTs, and 115 (38.6%) underwent ≥3 LRTs prior to LT. The patients with ≥3 LRTs had a significantly higher rate of bile leak compared to patients who received 1-2 LRTs (7.0 vs. 1.3%, *p* = 0.014). Unadjusted and adjusted regression analyses demonstrated a significant association between the total number of LRTs administered and bile leak, but not rates of overall biliary complications. The total number of LRTs was not significantly associated with any other peri-operative or post-operative outcome measure. These findings support the aggressive use of LRTs to control HCC in patients awaiting liver transplant, with further evaluation needed to confirm the biliary leak findings.

## 1. Introduction

Liver transplant has long been established as the gold-standard therapy for patients with localized hepatocellular carcinoma (HCC) not amenable to resection [1]. In 2002, the Milan Criteria were adopted by the United Network for Organ Sharing (UNOS), introducing exception points for HCC patients, and in 2015, the Organ Procurement and Transplantation Network (OPTN) implemented a policy change that required a mandatory wait time of 6 months prior to listing for a liver transplant. The average wait time for liver transplant varies widely by UNOS region, but for at least 65% of patients, the wait time exceeds 6 months, and for many patients, the wait time has gotten longer over time [2].

Local regional therapies (LRTs) such as transarterial chemoembolization (TACE), microwave ablation (MWA), and yitrium-90 (Y-90) radioembolization are effective treatments for HCC and have become increasingly necessary to prevent waitlist dropout and disease progression. The current clinical practice guidelines recommend using neoadjuvant LRTs in appropriately selected patients expected to wait more than 6 months for liver transplant [3], which means most patients undergoing transplant for HCC will receive at least one pre-operative LRT.

The LRTs are not without potential complications, and their impact on liver transplant outcomes is poorly understood. Previous studies have reported increased morbidity after resection following multiple LRTs [4], whereas others have found no differences in peri-operative outcomes [5]. In addition, hepatic arterial complications, including arteritis, pseudoaneurysms, stenosis, and thrombosis, have long been recognized as risks of TACE [6,7], but whether they lead to a higher prevalence of similar complications after transplantation remains unclear [8,9,10]. Similarly, TACE, MWA, and Y-90 have all been associated with bile duct ischemia and injury [11,12,13], but results have been mixed as to whether these therapies lead to an increased risk of biliary stricture or bile leak following transplant [8,14,15].

Given the increasing use of LRTs overall, particularly that of Y-90, new concerns have arisen over the impact these therapies may have on peri-operative outcomes [16,17,18]. This retrospective study used a contemporaneous cohort to evaluate the peri-operative outcomes and complication patterns associated with LRT use in patients with HCC awaiting liver transplant, with particular focus on those receiving multiple pre-operative LRTs or Y-90 radioembolization.

## 2. Materials and Methods

### 2.1. Study Design

This was a retrospective analysis of 298 consecutive patients with HCC who underwent liver transplant at a single academic comprehensive transplant center between May 2017 and May 2021. The data was collected retrospectively from two institution-specific databases and one national database containing quality metrics. The study was reviewed and approved by the institutional review board at the University of California, San Francisco, and was conducted in accordance with both the Declarations of Helsinki and Istanbul.

### 2.2. Patient Cohort

The cohort comprised adult patients diagnosed with HCC before transplant as well as a small number of patients diagnosed with HCC based on their explant pathology. The patients who underwent orthotopic liver transplants (OLT) or living donor liver transplants (LDLT) were included. The patients under the age of 18 were excluded. The data collected consisted of patient age, gender, race, etiology of liver disease, BMI, and date of initial HCC diagnosis. In addition, the baseline characteristics prior to transplant included pre-operative creatinine, total bilirubin, albumin, platelets, and alpha-fetoprotein (AFP), as well as the MELD score, the number and size of each HCC tumor, and whether patients were downstaged before transplant.

### 2.3. Liver-Directed Therapies

All decisions about the type of LRT to administer, the number of rounds, and the time interval between LRTs and other procedures were made by a multidisciplinary liver transplant team consisting of transplant surgeons, transplant hepatologists, oncologists, and interventional radiologists.

### 2.4. Peri-Operative Outcomes

The peri-operative outcomes data was collected prospectively by the UCSF Department of Anesthesia for Liver Transplantation. The peri-operative data included estimated blood loss (EBL), transfusion products received, including packed red blood cells (pRBCs), platelets, fresh frozen plasma (FFP), cryoprecipitate (cryo), and albumin, as well as urine output (UOP), case duration, cold ischemia time (CIT), and operative techniques used during the case. The peri-operative outcomes recorded included length of stay (LOS) and 30-day mortality. Additionally, the peri-operative complications included the need for reoperation, transfusion needs within 24 h, deep venous thrombosis (DVT), pulmonary embolism (PE), pneumonia (PNA), and urinary tract infection (UTI). Further, the post-transplant outcomes recorded included the need for readmission and post-transplant complications, including biliary, arterial, and venous complications, as well as any subsequent required interventions.

### 2.5. Transplant Eligibility

The patients with HCC were deemed eligible for liver transplant according to the UCSF Downstaging Criteria (UCSF-DS), which is defined as one lesion >5 cm and ≤8 cm, two or three lesions each less ≤5 cm, or four or five lesions each ≤3 cm, with a total tumor diameter of ≤8 cm. The patients were divided into three cohorts: those who met Milan criteria upfront; those who were within UCSF-DS criteria and were successfully downstaged to Milan criteria by the time of transplant; and those who were initially outside of UCSF-DS criteria and were successfully downstaged to Milan criteria (all comers).

### 2.6. Statistical Analyses

STATA/MP 17.0 and RStudio 1.4.1717 were used for all calculations. The categorical variables are reported as frequencies. The continuous variables are reported as medians and interquartile ranges. In addition, to test for differences by the total number of LRTs (0, 1–2, or 3+), chi-square tests were used for categorical variables, and Kruskal-Wallis tests were used for continuous variables. For continuous variables, the linear regression assumptions were tested with normal quantile plots and plotted with residuals against fitted values. Median regression was used if the assumptions for linear regression were not met; otherwise, linear regression was used. Further, for dichotomous variables, logistic regression was used. Logistic regression modeling was then used to examine the relationship between specific covariates and outcomes of interest, including arterial stenosis and biliary leak.

### 2.7. Propensity Score Matched Analysis

The patients with 3 or more LRTs (n = 115) were matched to patients with 1-2 LRTs (n = 156) using optimal full propensity score matching. A maximum of ten patients with 1–2 LRTs were matched with each patient with 3 or more LRTs. The matching was restricted to observations that had propensity scores in the extended common support region (0.13–0.71), which extends the common support region by 0.25 times a pooled estimate of the common standard deviation of the logit of the propensity score. Weighted matched standardized differences and variance ratios for the propensity score model covariates were used to assess sample balance after matching. Additionally, the acceptable balance was defined by a maximum of 0.1 for the absolute value of the standardized difference and by values within the 0.5–2 range for the variance ratio. In order to account for the matched nature of the sample, conditional logistic regression stratified on the matched pairs was performed. The PSMATCH procedure in SAS version 9.4 was used to perform optimal full propensity score matching.

## 3. Results

The study cohort consisted of 298 patients with HCC who underwent liver transplants at our center between May 2017 and May 2021. However, before undergoing transplant, 271 patients (90.9%) received at least one LRT, and 27 patients received none. As shown in Table 1, patients who received 0 LRTs were less likely to have multifocal disease and had HCC tumors that were significantly smaller than those of patients who received at least 1 LRT. The patients who received 0 LRTs prior to transplant had significantly higher median MELD scores prior to transplant than patients who had at least 1 LRT: 21 (IQR 17, 28) vs. 11 (IQR 8, 14) for patients who received 1–2 LRTs.

The breakdown of the number and type of LRTs received by this cohort is shown in Figure 1. The most common type of LRT used was TACE, with 216 patients (72.5%) receiving at least one TACE. The microwave ablation was the second-most common LRT, with 123 patients (41.3%) receiving at least one MWA. In addition, Y-90 radioembolization was the least common, with 47 patients (15.8%) receiving at least one Y-90 treatment. The median number of LRTs was two (IQR 1, 3), and the maximum number of LRTs received by a single patient was nine.

The peri-operative outcome measures are shown in Table 2 and are similarly stratified into three groups: patients who received 0 LRTs, 1-2 LRTs, and 3 or more LRTs. When compared with patients who received at least one LRT, patients with 0 LRTs prior to transplant had a significantly higher median EBL (5500 mL vs. 2000 mL) and significantly higher intra-operative transfusion requirements. The patients with 0 LRTs had significantly longer median case durations than those with either 1–2 LRTs or 3 or more LRTs: 539 min (IQR 480, 622) vs. 457 min (IQR 398, 535) vs. 447 min (IQR 386, 537). However, patients with 0 LRTs were also more likely than those who received at least one LRT to require alternative or complex biliary anastomoses such as Roux-en-Y hepaticojejunostomies (25.9% vs. 6.6%).

The patients who received 0 LRTs similarly fared worse on several post-operative outcome measures, as shown in Table 3. However, the patients who received 0 LRTs had longer LOS and higher transfusion requirements in the 24 h following transplantation. Their laboratory values on the day of discharge, including INR, total bilirubin, sodium, and creatinine, were all more abnormal than in patients who had received one or more LRTs. In addition, the patients with 0 LRTs were significantly more likely to require reoperation following their transplant, either during the same hospital stay or during a secondary admission. The HCC recurrence was observed in 6.9% of patients with a median time to recurrence of 0.9 years (IQR 0.6; suggested to delete blank lines 1.5) after LT. The overall recurrence-free survival rates for the entire cohort were 95.8% (95% CI 91.7suggested to delete blank lines 97.9) at 1-year and 92.5% (95% CI 86.9–95.8) at 3-years. There was no significant difference seen in HCC recurrence based on the type or number of local-regional treatments received.

These findings suggested that patients with HCC who received 0 LRTs prior to liver transplant were a distinct population from those who received one or more LRTs. In addition, patients who received no LRTs either had significant enough liver disease that they were not candidates for LRTs (n= 19) or were diagnosed incidentally on the basis of their post-operative pathology (n = 8). As a result, the patients who received 0 LRTs were excluded from all subsequent unadjusted and multivariable regression analyses.

In unadjusted analysis, only the total number of LRTs was significantly associated with bile leak. For each unit increase in LRT, the odds of a bile leak increased by 40% (OR 1.4, *p* = 0.03) and the need for a percutaneous drain increased by 45% (OR 1.45, *p* = 0.03). No other peri-operative or post-operative outcome measures were significantly associated with the total number of LRTs, as shown in Table 4.

The factors associated with post-operative biliary leak were further evaluated with unadjusted and adjusted logistic regression analyses. On unadjusted analysis, biliary leak was significantly associated with the total number of MWAs received (OR 2.06, *p* = 0.03) but was not significantly associated with the number of TACE or Y-90 treatments. The receipt of a living donor liver transplant (LDLT) was also significantly associated with post-operative bile leak (OR 5.46, *p* = 0.02). However, due to the fact that there were few biliary leaks, the logistic regression models with the total number of LRTs were adjusted for one covariate at a time, for a total of two independent variables in each model. The covariates associated with the risk of post-operative biliary complications were MELD score, BCLC stage, type of biliary anastomosis, arterial complications, and type of liver transplant (OLT vs. LDLT). Further, despite adjustment for these covariates, the total number of LRTs continued to be significantly associated with increased odds of bile leak (OR 1.36–1.46) (Table 5).

The distribution of LRTs administered to patients who developed post-transplant biliary leaks is shown in Figure 2. Six out of 10 patients with biliary leaks received at least 1 MWA prior to transplant. The prevalence of biliary leakage stratified by the total number of LRTs, TACEs, MWAs, and Y90s is additionally shown in Figure 2. The highest prevalence of biliary leak was seen in patients receiving 6 total LRTs (2 patients, 12.5%), 5 TACEs (1 patient, 25%), 3 MWAs (1 patient, 25%), and 2 Y90s (1 patient, 14.3%).

The factors associated with overall biliary complications are shown in Table 6. The unadjusted logistic regression demonstrated that increased odds of any biliary complications were significantly associated with the number of MWAs received (OR 1.46, *p* = 0.05), but not with the total number of LRTs. The increased odds of overall biliary complications were also significantly associated with MELD score (OR 1.08, *p* = 0.003) and the type of biliary anastomosis performed (Main Duct–Choledochcholedochostomy vs. Roux-en-Y Hepaticojejunostomy) (OR 4.5, *p* = 0.003). In a multivariable analysis that included the number of LRTs performed, the MELD score, the type of biliary anastomosis, and whether the patient underwent LDLT, both the MELD score and type of biliary anastomosis remained significantly associated with overall biliary complications.

A full propensity score matched analysis was performed to further evaluate the relationship between the number of LRTs and biliary and arterial complications following liver transplant. The propensity score model included recipient age at transplant, number of tumors at diagnosis, BMI, MELD, and graft type. All covariate data was complete in the study sample. The matched sample included 108 patients with 3 or more LRTs and 155 patients with 1–2 LRTs. Additionally, all covariates used in matching met sample balance criteria. In using conditional logistic regression stratified on the matched pairs, compared to patients with 1–2 LRTs, patients with 3 or more LRTs had higher odds of bile leak (OR 5.3, *p* = 0.03), did not have significantly different odds of biliary complications overall (*p* = 0.71), and had lower odds of arterial complications overall (OR 0.25, *p* = 0.04).

## 4. Discussion

The majority of patients who undergo liver transplantation for HCC will remain on the waitlist for more than 6 months, and more than 90% of these patients will receive at least one LRT prior to transplant [2]. The TACE, MWA, and Y-90 have all become invaluable tools in the prevention of tumor progression and waitlist dropout [2,19,20,21], but they also have well-documented complications that may become more likely with repeated use [9,16]. Our results suggest that increased use of preoperative LRTs, particularly TACE and MWA, may be associated with a higher risk of biliary leak post-transplant. No other peri-operative or post-operative measures included in this study were significantly associated with an increased number of LRTs.

The bile leaks and biliary strictures have long been recognized as complications of the TACE, MWA, and Y-90 [11,12,13]. The pathophysiology underlying these complications differs by the mechanism of action specific to each type of LRT, but typically results in biliary ischemia, which leads to stricture formation or leak. The arterially-based LRTs such as TACE or Y-90 can lead to mechanical trauma to involved arteries, occlusion of arteries and arterioles with lipiodol or gelfoam, as well as arteritis, all of which can, in turn, lead to biliary devascularization and ischemia [11,14,17]. In contrast, ablation procedures, such as MWA, can cause thermal damage to bile ducts, leading to fibrosis and necrosis [15].

In the absence of LRTs, biliary complications are common following liver transplantation, and several risk factors are well-described. These risk factors include hepatic arterial complications, older donor age, prolonged ischemia time, small duct size, complex biliary anatomy, acute rejection, and CMV infection [22]. In addition, biliary leak has been well-established to be more commonly associated with LDLT compared with OLT [23]. In our study, the incidence of biliary leak was significantly higher in patients who underwent LDLT, but even in this patient population, biliary leak continued to be associated with increased use of LRTs.

In our cohort, 19.5% of patients had biliary stricture, and 4.4% of patients had biliary leaks. Both proportions are similar to those in previous reports [8,14]. After controlling for known confounders, biliary leakage continued to be significantly associated with the total number of LRTs received, but overall biliary complications were not. The reason for this is not immediately clear, but one possibility is that the risk factors for biliary leak and biliary stricture are not exactly the same. The bile leaks tend to present earlier, often within 1 month of transplant, than strictures, which can form up to 6 months after transplant. Biliary leaks are most commonly associated with technical or ischemic complications of the bile duct, whereas strictures can also be secondary to duct size, donor factors including age and hypotension, immunologic injury (rejection, recurrent autoimmune disease), and infectious complications such as CMV [24]. We hypothesize that LRT use may be associated with localized ischemia of the biliary system or possibly, increased technical complexity of the transplant itself, but we would expect no association with donor-related risk factors, or longer-term immunologic or infectious complications.

The association with biliary leaks was primarily driven by the number of TACE and MWA procedures. The number of MWA procedures was significantly associated with biliary leak (OR 2.1, *p* = 0.03), but the association with the number of TACE procedures was not statistically significant (OR 1.2, *p* = 0.28). Prior studies of liver transplant outcomes following TACE have not demonstrated a significantly increased risk of biliary complications [8,10,14]. These studies did not evaluate the number of TACE procedures performed, used heterogenous comparator patient groups, and patients who received zero LRTs were not excluded. A few studies have reported on liver transplant outcomes following MWA, and we know of no study that reported specifically on biliary complications. Our results suggest that the use of MWA was associated with higher odds of biliary leak than either TACE or Y90, but further validation of this association is required to determine whether it should be considered when evaluating which LRT to administer pre-transplant.

In contrast to biliary complications, hepatic artery complications have been previously associated with LRT use pre-transplant [6] and have been of particular concern for arterially-delivered therapies such as TACE and Y90. A dual-center retrospective cohort study published by Goel et al. in 2014 demonstrated a higher risk of hepatic arterial stenosis in patients who received TACE [8]. A higher risk of arterial stenosis following TACE was also reported in a 2018 meta-analysis [10], but was not supported by a subsequent multicenter cohort study [9]. A few studies have evaluated post-transplant outcomes in patients who received Y-90, but one small cohort study reported no association between Y-90 use and arterial complications [18]. Our own results demonstrated no association between the risk of arterial complications and the total number of LRTs, nor any association with the use of TACE, MWA, or Y-90.

Our results were also reassuring about the effect that LRT use might have on overall operative difficulty. LRT use, particularly MWA, can cause complications such as adhesion formation, diaphragm necrosis and rupture, abscesses, and needle track metastases, all of which could significantly complicate a liver transplant [5]. Prior studies have had mixed results as to whether these complications lead to longer operative times, increased transfusion requirements, and higher rates of post-operative complications overall [4,5]. Our findings, however, suggest that increased use of LRTs pre-operatively is not associated with case duration, EBL, transfusion requirements, or any other outcome measure. Similarly, the use of pre-operative Y-90 was not significantly associated with any post-operative outcome measures.

There are several limitations to acknowledge for this study. First, as a retrospective cohort study, it cannot establish a clear causal relationship between LRT use and our outcomes of interest. Although many potential confounders were included in our multivariable models, there may be other predictors that were not measured. Second, this study was based on data from a single center, which limits the generalizability of our results to centers that may use different algorithms to treat their HCC patients. Third, the overall prevalence of biliary complications, particularly biliary leaks, was small, which may have made it difficult to detect small differences, as well as to control for all possible confounders without overfitting our multivariable model. Lastly, given the recent nature of our data, we were unable to report on long-term survival and post-transplant HCC recurrence, which would also be important to consider in optimizing patterns of LRT use.

## 5. Conclusions

In summary, our results suggest that increased use of LRTs prior to a liver transplant may lead to a higher risk of post-operative bile leak. Although bile leaks are an important complication to consider, they are typically treatable with endoscopic or percutaneous interventions and will generally resolve with time. The findings of our study, however, suggest that evaluating for evidence of duct damage or ischemia using a technology such as ICG may be warranted in patients who have received multiple LRTs prior to transplant [25]. However, should duct damage be identified, the use of a Roux-en-Y hepaticojejunostomy or other alternative method of biliary reconstruction should be considered.

Further work is needed to confirm our findings and better elucidate which patients are at the highest risk. Reassuringly, no other post-operative outcomes were found to be significantly associated with the number of LRTs administered, which suggests that LRTs should continue to be liberally used in patients with localized HCC awaiting liver transplant.

## Figures and Tables

**Figure 1 cancers-15-00620-f001:**
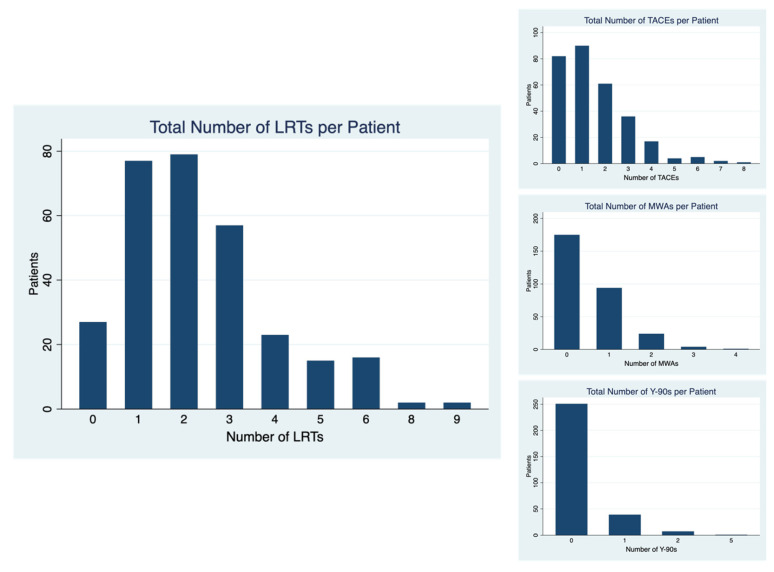
Number and Type of LRTs per Patient. LRT = Local Regional Therapy, TACE = Transarterial Chemoembolization, MWA = Microwave Ablation, Y-90 = Yttrium-90 Radioembolization.

**Figure 2 cancers-15-00620-f002:**
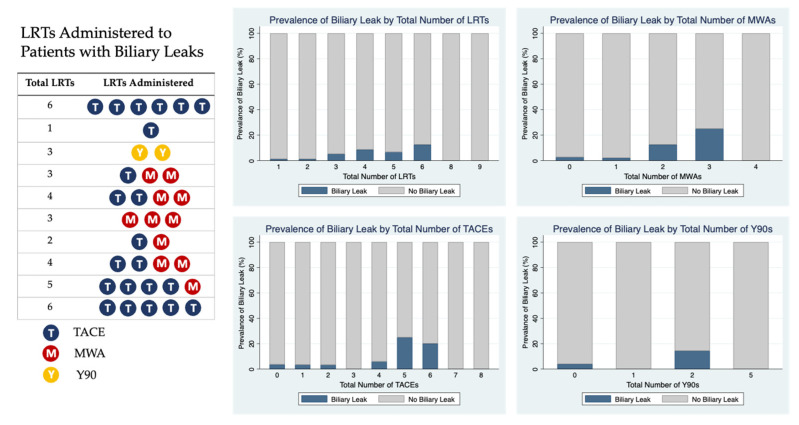
Prevalence of Biliary Leak for Different Type of TRL.

**Table 1 cancers-15-00620-t001:** Demographic and Clinical Characteristics of Liver Transplant Recipients.

Number of LRTs	0 (n = 27)	1–2 (n = 156)	3+ (n = 115)	*p*-Value
Age (median, IQR)	59 (56, 64)	63 (58, 67)	64 (61, 68)	0.007 *
Gender				
Male (%)	19 (70.4%)	125 (80.1%)	85 (73.9%)	0.34
Female (%)	8 (29.6%)	31 (19.9%)	30 (26.1%)	
Etiology of Liver Disease				
HCV (%)	6 (22.2%)	76 (48.7%)	68 (59.1%)	0.002 *
HBV (%)	3 (11.1%)	25 (16.0%)	18 (15.7%)	0.81
ETOH (%)	9 (33.3%)	28 (17.9%)	16 (13.9%)	0.06
NASH (%)	8 (29.6%)	25 (16.0%)	17 (14.8%)	0.17
Other	3 (11.1%)	13 (8.3%)	10 (8.7%)	0.89
HCC at Diagnosis				
Number of tumors (median, IQR)	1 (0, 1)	1 (1, 1)	1 (1, 2)	0.0003 *
Size of largest tumor (cm) (median, IQR)	2 (0, 2.7)	2.4 (2.1, 3)	2.7 (2, 3.9)	0.0004 *
HCC Diagnostic Criteria				
No HCC diagnosed prior to LT (%)	8 (28.6%)	0 (0%)	1 (0.9%)	<0.0001 *
Within Milan (%)	17 (63.0%)	149 (95.5%)	82 (71.3%)	
Downstaging Criteria (%)	1 (3.7%)	5 (3.2%)	23 (20.0%)	
All-Comers (%)	1 (3.7%)	2 (1.3%)	9 (7.8%)	
BMI (median, IQR)	28.0 (25.1, 31.4)	28.6 (25.5, 32.1)	26.6 (23.7, 31.0)	0.09
Labs at Time of Listing				
MELD (median, IQR)	21 (17, 28)	11 (8, 14)	10 (7, 12)	0.0001 *
Total Bilirubin (median, IQR)	4.7 (2.4, 5.8)	1.4 (0.8, 2.5)	1.2 (0.8, 1.7)	0.0001 *
Albumin (median, IQR)	2.6 (2.2, 3.0)	3.5 (3.4, 3.6)	3.5 (3.3, 3.6)	0.16
Platelets (median, IQR)	66 (54, 79)	91 (59, 138)	86 (60.5, 153.5)	0.20
AFP (median, IQR)	6.8 (3.3, 31.4)	5.1 (2.9, 10.9)	9.6 (4.7, 28.4)	0.0007 *
Graft Type				
OLT (%)	21 (77.8%)	141 (90.4%)	108 (93.9%)	0.04
LDLT (%)	6 (22.2%)	15 (9.6%)	7 (6.1%)	

* *p*-value < 0.05. LRT = Local Regional Therapy, CI = Confidence Interval, IQR = Interquartile Range, HCV = Hepatitis C Cirrhosis, HBV = Hepatitis B Cirrhosis, ETOH = Alcohol-Induced Cirrhosis, NASH = Non-alcoholic Steatohepatitis, HCC = Hepatocellular Carcinoma, BMI = Body Mass Index, MELD = Model for End-Stage Liver Disease, AFP = Alpha-Feto Protein, OLT = Orthotopic Liver Transplantation, LDLT = Living Donor Liver Transplantation

**Table 2 cancers-15-00620-t002:** Peri-operative Measures Stratified by Total Number of LRTs.

Number of LRTs	0 (n = 27)	1–2 (n = 156)	3+ (n = 115)	*p*-Value
EBL (mL) (median, IQR)	5500 (2500, 9400)	2000 (1000, 4050)	2000 (1000, 3250)	0.0001 *
pRBCs (units) (median, IQR)	7 (3, 10)	1 (0, 4)	2 (0, 4)	0.0001 *
FFP (units) (median, IQR)	14 (7, 21)	4.5 (1.5, 13)	5 (2, 9)	0.0001 *
Platelets (units) (median, IQR)	3 (1, 4)	1 (0, 3)	1 (0, 2)	0.0001 *
Crystalloid (mean, 95% CI)	1478 (1167, 1788)	1585 (1463, 1708)	1565 (1417, 1713)	0.81
Albumin (ml) (median, IQR)	1000 (0, 1500)	1500 (625, 2000)	1576 (1405, 1746)	0.04 *
Urine Output (ml) (median, IQR)	800 (250, 1200)	705 (450, 1075)	650 (375, 1100)	0.79
Extubation in the OR (%)	16 (59.3%)	115 (73.7%)	85 (73.9%)	0.15
Case Duration (minutes) (median, IQR)	539 (480, 622)	457 (398, 535)	447 (386, 537)	0.02 *
Cold Ischemia Time (hours) (median, IQR)	6.2 (4.0, 8.0)	7.4 (5.8, 9.0)	7.0 (6.0, 8.8)	0.16
Biliary Anastomosis (% Complex or Roux-en-Y HJ)	7 (25.9%)	12 (7.7%)	6 (5.2%)	0.02 *
Venous Outflow (% Bicaval)	3 (11.1%)	13 (8.3%)	13 (11.3%)	0.69

* *p*-value < 0.05. LRT = Local Regional Therapy, CI = Confidence Interval, IQR = Interquartile Range, EBL = Estimated Blood Loss, pRBC = Packed Red Blood Cells, FFP = Fresh Frozen Plasma, OR = Operating Room, R-en Y HJ = Roux-en-Y hepaticojejunostomy

**Table 3 cancers-15-00620-t003:** Post-operative Measures Stratified by Total Number of LRTs.

Number of LRTs	0 (n = 27)	1–2 (n = 156)	3+ (n = 115)	*p*-Value
Hospital Stay				
Overall LOS (days) (median, IQR)	9.5 (6, 13)	7 (5, 9)	7 (5, 9)	0.01 *
ICU LOS (days) (median, IQR)	2 (1, 4)	2 (1, 3)	2 (1, 2)	0.14
pRBCs within 24 h (units) (median, IQR)	7 (3, 10)	1 (0, 4.5)	1 (0, 3)	0.0001 *
Need for reoperation (same hospital stay)	4 (14.8%)	13 (8.3%)	11 (9.6%)	0.57
Labs at time of Discharge				
Final INR (median, IQR)	1.9 (1.3, 2.8)	1.4 (1.2, 1.8)	1.3 (1.1, 1.6)	0.0001 *
Final Bilirubin (median, IQR)	5.6 (2.8, 19.6)	1. (0.9, 2.9)	1.5 (0.8, 2.1)	0.0001 *
Final Sodium (median, IQR)	134 (131, 137)	137 (137, 138)	137 (136, 139)	0.0009 *
Final Creatinine (median, IQR)	1.1 (0.9, 1.4)	0.9 (0.8, 1.1)	0.9 (0.7, 1.1)	0.03*
Complications				
Organ Space SSI	1 (3.7%)	2 (1.9%)	3 (2.6%)	0.83
Wound Dehiscence	0	0	1 (0.9%)	0.45
Number of Readmissions (median, IQR)	1 (0, 3)	0 (0, 1)	0 (0, 1)	0.13
Reoperation (overall)	8 (29.6%)	16 (10.3%)	14 (12.2%)	0.02 *
Biliary Stricture	8 (29.6%)	30 (19.2%)	20 (17.4%)	0.35
Biliary Leak	3 (11.1%)	2 (1.3%)	8 (7.0%)	0.02 *
Biliary Complication requiring ERCP	9 (33.3%)	30 (19.2%)	24 (20.9%)	0.25
Biliary Leak requiring IR Drain	2 (7.4%)	2 (1.3%)	6 (5.2%)	0.10
Arterial Stenosis	0	11 (7.1%)	3 (2.6%)	0.11
Arterial Stenosis requiring Angioplasty	0	9 (5.8%)	3 (2.6%)	0.23
Venous Stenosis	2 (7.4%)	7 (4.5%)	3 (2.6%)	0.48
Venous Stenosis requiring Angioplasty	1 (3.7%)	7 (4.5%)	2 (1.7%)	0.46

* *p*-value < 0.05. LRT = Local Regional Therapy, IQR = Interquartile Range, LOS = Length of Stay, ICU = Intensive Care Unit, pRBCs = Packed Red Blood Cells, INR = International Normalized Ratio, SSI = Surgical Site Infection, ERCP = Endoscopic Retrograde Cholangiopancreatography, IR = Interventional Radiology.

**Table 4 cancers-15-00620-t004:** Unadjusted Regression Models for Outcomes vs. Total Number of LRTs (Continuous).

Linear or Median Regression Outcome	Coefficient	95% CI	*p*-Value
Peri-operative			
EBL (ml) *	83.3	(−71.4, 238.0)	0.29
pRBCs (units) *	0.25	(−0.12, 0.62)	0.19
FFP (units) *	0	(−0.57, 0.57)	>0.99
Platelets (units) *	1.74	(−2.51, 5.99)	0.42
Case Duration (minutes) *	−2.03	(−16.6, 12.6)	0.79
Cold Ischemia Time (hours) *	−0.78	(−2.5, 0.9)	0.37
Hospital Stay			
Overall Length of Stay (days) *	0	(−0.26, 0.26)	>0.99
ICU Length of Stay (days)	0	(−0.12, 0.12)	>0.99
pRBCs within 24 h (units)	0	(−0.25, 0.25)	>0.99
Complications			
Number of Readmissions	−0.04	(−0.13, 0.04)	0.34
Total Subsequent Operations & Procedures	0.08	(−0.08, 0.24)	0.30
Logistic Regression Outcome	OR	95% CI	*p*-Value
Anatomic			
Biliary Anastomosis (Main Duct-choledocholedochostomy vs. Complex or Roux-en-Y HJ)	1.01	(0.75, 1.36)	0.96
Venous Outflow (Piggyback vs. Bicaval)	1.20	(0.96, 1.50)	0.11
Complications			
Reoperation (overall)	1.06	(0.84, 1.33)	0.61
Biliary Stricture	1.04	(0.86, 1.26)	0.67
Biliary Leak	1.4	(1.03, 1.90)	0.03 **
Biliary Complication requiring ERCP	1.11	(0.93, 1.32)	0.26
Biliary Leak requiring IR Drain	1.45	(1.04, 2.02)	0.03 **
Arterial Stenosis	0.72	(0.46, 1.14)	0.16
Arterial Stenosis requiring angioplasty	0.72	(0.44, 1.18)	0.19
Venous Stenosis	0.92	(0.60, 1.41)	0.70
Venous Stenosis requiring angioplasty	0.72	(0.41, 1.27)	0.26

* Linear regression assumptions not met. Median regression used. ** *p*-value < 0.05. CI = Confidence Interval, OR = Odds Ratio, EBL = Estimated Blood Loss, pRBC = Packed Red Blood Cells, FFP = Fresh Frozen Plasma, ICU = Intensive Care Unit, R-en Y HJ = Roux-en-Y Hepaticojejunostomy, ERCP = Endoscopic Retrograde Cholangiopancreatography, IR = Interventional Radiology.

**Table 5 cancers-15-00620-t005:** Logistic Regression Models for Biliary Leak.

**Unadjusted Logistic Regression Models for Biliary Leak vs. Risk Factor**			
**Risk Factor**	**OR**	**95% CI**	***p*-Value**
Total Number of LRTs	1.40	(1.03, 1.90)	0.03 *
Total TACEs	1.22	(0.85, 1.76)	0.28
Total MWAs	2.06	(1.06, 3.99)	0.03 *
Total Y-90s	0.95	(0.28, 3.23)	0.93
Age	1.00	(0.91, 1.10)	0.93
Donor Age	0.95	(0.91, 0.99)	0.03 *
MELD	1.04	(0.94, 1.15)	0.42
Cold Ischemia Time	0.95	(0.75, 1.21)	0.69
Case Duration	1.00	(0.99, 1.00)	0.07
Estimated Blood Loss	1.00	(1.00, 1.00)	0.70
Intra-op pRBCs (units)	1.03	(0.96, 1.11)	0.37
Arterial Stenosis	2.12	(0.25, 18.0)	0.49
LDLT	5.46	(1.31, 22.8)	0.02 *
Biliary Anastomosis (Main Duct-choledochole-dochostomy vs. Complex or Roux-en-Y HJ)	3.83	(0.75, 19.5)	0.11
**Logistic Regression Models for Biliary Leak vs. Total Number of LRTs, Controlling for Potential Confounders**			
**Risk Factor**	**OR**	**95% CI**	***p*-Value**
Total Number of LRTs	1.43	(1.04, 1.96)	0.03 *
Biliary Anastomosis (Main Duct-choledochole-dochostomy vs. Complex or Roux-en-Y HJ)	4.23	(0.80, 22.3)	0.09
Total Number of LRTs	1.44	(1.05, 1.98)	0.02 *
LDLT status	6.41	(1.46, 28.2)	0.01 *
Total Number of LRTs	1.43	(1.04, 1.96)	0.03 *
Arterial Stenosis	3.07	(0.34, 27.8)	0.32
Total Number of LRTs	1.46	(1.06, 2.02)	0.02 *
MELD	1.07	(0.96, 1.19)	0.22
Total Number of LRTs	1.36	(1.00, 1.86)	0.05 *
Donor Age	0.96	(0.92, 1.00)	0.04 *
Total Number of LRTs	1.44	(1.04, 1.97)	0.03 *
Cold Ischemia Time	0.93	(0.73, 1.19)	0.57
Total Number of LRTs	1.36	(1.00, 1.86)	0.02 *
Case Duration	1.00	(1.00, 1.00)	0.05 *

* *p*-value < 0.05. CI = Confidence Interval, OR = Odds Ratio, LRT = Local Regional Therapy, TACE = Transarterial Chemoembolization, MWA = Microwave Ablation, Y-90 = Yttrium-90 Radioembolization, MELD = Model for End-Stage Liver Disease, pRBCs = Packed Red Blood Cells, LDLT = Living Donor Liver Transplantation, R-en Y HJ = Roux-en-Y Hepaticojejunostomy.

**Table 6 cancers-15-00620-t006:** Logistic Regression Models for Any Biliary Complications.

**Unadjusted Logistic Regression Models for Any Biliary Complications vs. Risk Factor**			
**Risk Factor**	**OR**	**95% CI**	***p*-Value**
Total Number of LRTs	1.10	(0.93, 1.32)	0.25
Total TACEs	1.06	(0.87, 1.29)	0.56
Total MWAs	1.46	(1.00, 2.12)	0.05 *
Total Y-90s	0.86	(0.47, 1.57)	0.62
Age	0.99	(0.95, 1.03)	0.59
Donor Age	0.99	(0.97, 1.01)	0.19
MELD	1.08	(1.03, 1.14)	0.003 *
Cold Ischemia Time	0.99	(0.90, 1.08)	0.75
Case Duration	1.00	(1.00, 1.00)	0.43
Estimated Blood Loss	1.00	(1.00, 1.00)	0.42
Intra-op pRBCs (units)	1.03	(0.99, 1.08)	0.19
Arterial Stenosis	1.62	(0.49, 5.36)	0.43
LDLT	2.46	(0.97, 6.19)	0.06
Biliary Anastomosis (Main Duct-choledochole-dochostomy vs. Complex or Roux-en-Y HJ)	4.50	(1.69, 11.96)	0.003 *
**Multivariable Logistic Regression Models for Any Biliary Complications**			
**Risk Factor**	**OR**	**95% CI**	***p*-Value**
Total Number of LRTs	1.17	(0.97, 1.42)	0.09
Biliary Anastomosis (Main Duct-choledochole-dochostomy vs. Complex or Roux-en-Y HJ)	4.38	(1.42, 13.5)	0.01 *
MELD	1.09	(1.04, 1.16)	0.001 *
LDLT	1.33	(0.44, 4.06)	0.62
Donor Age	0.99	(0.97, 1.01)	0.29

* *p*-value < 0.05. CI = confidence interval, OR = odds ratio, LRT = local regional therapy, TACE = transarterial chemoembolization, MWA = microwave ablation, Y-90 = Yttrium-90 radioembolization, MELD = Model for End-Stage Liver Disease, pRBCs = Packed Red Blood Cells, LDLT = living donor liver transplantation, R-en Y HJ = Roux-en-Y hepaticojejunostomy.

## Data Availability

The data presented in this study are available on request from the corresponding author. The data are not publicly available due to privacy restrictions.

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
