# Peer review of "Number of Local Regional Therapies for Hepatocellular Carcinoma and Peri-Operative Outcomes after Liver Transplantation"

_cancers, 2023, doi:10.3390/cancers15030620_

Round 1
Reviewer 1 Report
In this study, the authors evaluted the effect of multiple rounds of loco-regional therapies (LRTs) in hepatocellular carcinoma (HCC) patients who underwent liver transplant (LT). They discovered a significant association between the total number of LRTs and a higher rate of post-operative bile leak. However, there was no association between the number of LRTs and other peri- or post-operative outcomes. Although a large number of HCC patients underwent LT were evaluated, this paper has several limitations.
Major comments:
- As documented by authors, patients recieved no LRT exhibited unique baseline characterstics. Therefore, authors need to compensate the difference of baseline characteristics by performing a propensity score matched analysis. Nontheless, excluding the patients with 0 LRT at baseline and performing further analyses may help to determine which variable influences the peri- or post-operative complications.
- The total number of biliary leakage cases is just ten, making it very susceptible to bias. Due to the small number of cases, it is difficult to conclude that the total number of LRTs is an independnent factor. Multivariable analysis or, if difficult, further subgruop analysis for additional variables that may influence biliary leakage should be performed (e.g. variables known as risk factors for biliary compliations in previous studies, such as transfusion, operation time, and ischemic time).
- In the Dicussion section, authors explained that biliary leakage increased as a result of artery or bile duct injury after LRT; however, more explanation is required as to why other biliary complications did not increased.
- Thermal ablation with radiofrequency (RFA) is the standard of care for patients with early stage of HCC. In this study, there are patients underwent microwave ablation (MWA) rather than RFA were documented. Was the ablation performed only by microwave?
- As the study was conducted on HCC patients, please share the HCC recurrence outcomes.
Reviewer 2 Report
The authors evaluated peri-operative outcomes and complication patterns associated with LRTs in patients with HCC awaiting liver transplant and revealed that total number of LRTs and total MWAs are associated with bile leak, but the association with peri-operative and post-operative outcome measure was not significant.
I concern that there are several issues for publication.
1. The authors should show the prevalence of the biliary complications according to receiving MWA/no MWA, TACE/no TACE, and Y90/no Y90. In addition, please present the prevalence of the biliary complication according to the number of LRTs.
2. The authors revealed that the number of MWA procedures was associated with biliary leak, but the association with the number of TACE and Y90 procedures was not significant. Do these findings suggest that TACE or Y90 rather than MWA is recommended therapies as the LRT in patients with HCC awaiting liver transplant? This point should be discussed.
3. In this cohort, the main LRT is TACE procedure (216 pts, 72.5%) and total number of LRTs is associated with biliary leak. From these findings, I guess that many patients received repeated TACE; however, the association with the number of TACE was not significant. Please show what kind of therapy and how many therapy the patients with biliary leak received.
Round 2
Reviewer 1 Report
Thank you for your revision.
I think the manuscript is revised well.
Reviewer 2 Report
The authors adequately revised the manuscript according to the reviewer’s comments.